# rGO-PDMS Flexible Sensors Enabled Survival Decision System for Live Oysters

**DOI:** 10.3390/s23031308

**Published:** 2023-01-23

**Authors:** Pengfei Liu, Luwei Zhang, You Li, Huanhuan Feng, Xiaoshuan Zhang, Mengjie Zhang

**Affiliations:** Beijing Laboratory of Food Quality and Safety, College of Engineering, China Agricultural University, Beijing 100083, China

**Keywords:** shell-closing strength, flexible sensor, survival prediction, live oysters

## Abstract

The shell-closing strength (SCS) of oysters is the main parameter for physiological activities. The aim of this study was to evaluate the applicability of SCS as an indicator of live oyster health. This study developed a flexible pressure sensor system with polydimethylsiloxane (PDMS) as the substrate and reduced graphene oxide (rGO) as the sensitive layer to monitor SCS in live oysters (rGO-PDMS). In the experiment, oysters of superior, medium and inferior grades were selected as research objects, and the change characteristics of SCS were monitored at 4 °C and 25 °C. At the same time, the time series model was used to predict the survival rate of live oyster on the basis of changes in their SCS characteristics. The survival times of superior, medium and inferior oysters at 4 °C and 25 °C were 31/25/18 days and 12/10/7 days, respectively, and the best prediction accuracies for survival rate were 89.32%/82.17%/79.19%. The results indicate that SCS is a key physiological indicator of oyster survival. The dynamic monitoring of oyster vitality by means of flexible pressure sensors is an important means of improving oyster survival rate. Superior oysters have a higher survival rate in low-temperature environments, and our method can provide effective and reliable survival prediction and management for the oyster industry.

## 1. Introduction

Oysters are one of the most popular seashell foods in the world due to their high nutritional value and delicious taste [1,2,3,4,5]. Thanks to a coastal line stretching 18,000 km, China also has abundant oyster resources and a long history of their utilization [6]. China accounts for most of the world’s oyster aquaculture production, representing 83% of the production by weight in 2019 [7]. Oysters are consumed in many forms around the world, in both raw and cooked forms [8,9]. Compared with cooked oysters, raw oyster consumption is more popular due to the fresh taste and high nutritional value of oysters [10]. Oysters are susceptible to acute damage or death from environmental stress. The abundant nutrients in oysters provide a good external environment for the growth and reproduction of microorganisms [11]. Dead oysters will deteriorate under the action of digestive enzymes and oxidation, producing a pungent odor [12]. Therefore, effective indicators and strategies should be adopted for dynamic monitoring to improve the survivability of live oysters.

The adductor muscle plays an important physiological role, which includes opening and closing the two shells, as well as being an energy reserve for maintaining normal conditions [13]. Healthy oysters generally close their valves tightly in response to various stress. Therefore, the strength of the adductor muscle in the oyster may reflect its health status. In practice, consumers usually judge whether an oyster is dead by observing the opening and closing of the shell. In previous studies, the adductor muscle has been shown to control the opening and closing of the shell, and its structural characteristics and mechanical response may be useful indicators reflecting the health status of shellfish. Previous studies have shown the validity of measuring adductor muscle strength as an indicator for diagnosing the health of oysters [14]. However, these detection devices are usually inflexible and destructive, resulting in damage or death to live oysters.

In recent years, many nondestructive methods have found wide application for the quality and safety evaluation of marine products, including near-hyperspectral imaging and computer vision. Computer vision has been used to analyze digital images in order to perform rapid visual evaluations, on the basis of which spatial information regarding the tested samples, including size, shape, color, and texture, can be obtained [15,16,17,18]. Hyperspectral imaging provides the physical and geometric characteristics, such as shape, size, appearance and color, as well as the chemical composition of the analyzed sample, through spectral analysis [19,20]. However, unlike seafood other than oysters, the two aforementioned methods are not able to penetrate the oyster shell, and are therefore usually only used to measure the freshness of the oyster meat. Therefore, there is an urgent need to discover a new method for monitoring the health of live oysters during storage and transportation.

As an effective and promising technology, flexible sensing technology has shown great advantages in the measurement of physiological indicators such as heart rate, blood sugar and lactic acid [21,22,23]. In recent years, flexible sensing technology has found gradual application in the health monitoring of seafood, including the measurement of compounds and pH values. In addition, PDMS-coated laser-induced graphene bending sensors were integrated with ultra-low-power aquatic tags and utilized in underwater animal speed monitoring applications [24]. On the one hand, all of these studies demonstrated the feasibility and potential of flexible sensing technology in the quality and health monitoring of seafood. On the other hand, there are few applications based on flexible sensing technology in shellfish physiological monitoring. Currently, piezoresistive pressure sensors are being widely developed and applied because of their obvious advantages, which include a simple fabrication process, high sensitivity, high resolution, fast response, good scalability and wide detection range [25,26,27]. The sensor was successfully applied for not only detecting human motions, including radial pulse, speech recognitions and joint movements, but also for monitoring health status [28]. However, these sensors have not been effectively applied in the monitoring of seafood quality.

It has been indicated in previous reports in the literature that closure strength was the most important index for evaluating the health status of oysters [29]. Conventional sensors can cause damage to live oysters during monitoring, affecting their survival time. In contrast, flexible pressure sensors can not only effectively monitor live oyster SCS, but they also do not affect their health. Therefore, the use of flexible pressure sensors to monitor the closed shell strengths of oysters is reliable and promising. The aim of this study was to develop a basic rGO/PDMS piezoresistive pressure sensor system and to explore the potential of flexible pressure sensors for the monitoring of oyster health and quality detection.

## 2. Materials and Methods

### 2.1. Overall System Framework

The intelligent system for oyster health evaluation consists of three architectural layers: the data acquisition layer, the management layer, and the application layer, as shown in Figure 1. System design and development are performed using the following three steps:
(1)Determine the mechanism for the evaluation of oyster survival and the health and comfort of the environment, and select oyster SCS as the physiological monitoring indicator.(2)Design a physiological signal and environmental information collection solution based around wearable and non-destructive flexible pressure sensors.(3)Establish a dynamic health and survival evaluation model and develop the corresponding decision-making support system.

### 2.2. Sensor Fabrication

Figure 2 illustrates the fabrication process and characterization of the flexible rGO-PDMS pressure sensor, which is composed of a PDMS substrate and an rGO sensitive layer. The fabrication steps of the PDMS substrate and the spin-coated rGO sensitive layer are described in Figure 2. Polydimethylsiloxane (PDMS) thermal membranes and other materials were purchased from Shang Mifang Electronic Technology (Shanghai, China) Co., Ltd. Firstly, PDMS pre-polymer (base) and cross-linker (curing agent) were mixed in a ratio of 10:1 and then placed in a vacuum chamber for 10 min to remove bubbles. PDMS was spin-coated (15 min at 1000 rpm) onto the glass slide surface and then cured at 80 °C for 120 min. The dried PDMS film was plasma processed for 10 min to enhance its hydrophilicity. Next, the PDMS surface was spin-coated with rGO solution and dried in an oven (80 °C), then repeatedly spin-coated 10 times with 5% rGO solution of N,N-dimethylformamide (5 min at 600 rpm). The PDMS film was then peeled off the slide and the two rGO-coated films were assembled face to face. Finally, silver electrodes were attached to the rGO sensing layer with silver adhesive and encapsulated. The overall structure of the completed flexible pressure sensor is shown in Figure 2.

### 2.3. System Function Design

#### 2.3.1. Hardware Design and Signal Processing

In this paper, the flexible pressure sensor device is mainly designed for the purpose of being able to adequately perform oyster SCS monitoring. The rGO-PDMS system mainly consists of a power supply module, a signal conditioning circuit, a main controller, a communication module, a sensing element, and an upper computer. In order to realize long-distance transmission while reducing costs, the transmission method includes serial communication and wireless communication. The real-time resistance of the sensor is calculated by the operational circuit of the operational amplifier. As shown in Figure 3a, the output resistance is calculated on the basis of the reference voltage, the output voltage, and the feedback resistance. The calculation formula is as follows:(1)$$R_{Sensor}=\left(\frac{V_{REF}}{V_{OUT}}\right)*R_{\mathrm{feedback}}$$

The formula $R_{Sensor}$ is the real-time resistance of the sensor, $V_{REF}$ is the reference voltage, $V_{OUT}$ is the output voltage, and $R_{\mathrm{feedback}}$ is the feedback resistance.

The serial communication connects the system and PC by means of Micro USB, which is able to read the pressure value and record the real-time data, and the output file type is xlsx. Bluetooth communication is used to transfer the data to a cell phone for remote detection. The controller is based on the high-performance ATMEGA32U4 chip operating at 5 V/16 MHz. The controller provides four 10-bit ADC pins, 12 digital I/O, and hardware serial connections to Rx and Tx. An optional 0.96-inch (4-pin) LCD screen can be used to display the sensor monitoring data in real time. The signal conditioning circuit converts the analog signal from the resistor into a digital signal and performs signal amplification, filtering, and denoising.

#### 2.3.2. Software Design

The software of the wireless sensor network was designed on the basis of the implementation of two main functions: (1) dynamic transmission and collection of environmental information and physiological signals; and (2) feedback and control of the transport. The main controller executes system commands including system initialization and interrupts, sensor initialization, Bluetooth module initialization, data writing and sending, controller restart, APP connection request, etc. The implementation of the decision support system is shown in Figure 4. The system monitoring interface includes data acquisition, curve generation, historical data query, storage management and data analysis. When the initialization command fails, the LED indicator flashes, data cannot be transmitted, and the controller repeats the initialization command. When the initialization command is successful, the LED indicator is always on and the data can be transmitted. When sending data fails, the system re-executes the command to send data. When the environment becomes abnormal during transportation, the software interface displays the abnormality and a warning. At this time, the platform sends commands to the transport end controller and the transport personnel to ensure normal operation. The wireless sensor network performs data collection and transmission during transportation. After transport is completed, the system automatically enters low-power mode. The software system monitoring interface is shown in Figure 5.

### 2.4. Data Analysis

As a physiological indicator of live oysters, SCS is the most important factor affecting their survival rate. When given the transport time or distance, mathematical models can be constructed to dynamically predict the survival rate of the terminal. Survival times are arranged according to a predetermined time interval (h), and survival rates are predicted using a time series model with one exponential smoothing method. The smooth recursive relationship of the survival rate was calculated with Equation (2), as follows:(2)$$Survival_{pre|i}=\delta Survival_{act|i}+\left(1-\delta \right)Survival_{rate|i-1}$$

$Survival_{rate|i}$ is the smoothed value at time node i, and $Survival_{act|i}$ is the actual survival rate at this time node. δ can be any value between 0 and 1, and it controls the balance between the survival rate at the previous time node and the next node: when δ is close to 1, only the current survival rate data point is retained; when δ is close to 0, only the previous smoothed value is retained. The detailed recursive relational equation is as follows:(3)$$Survival_{ratei}=\delta \overset{i}{\underset{j=0}{\sum }}\left(1-\delta \right)Survival_{act|i-j}$$

### 2.5. Experimental Scenario and Process Design

The experimental design consists of two parts: the sensor electrochemical performance test and oyster SCS monitoring. Throughout the entire experimental process, the sensor was operated in a high-frequency, high-strength, and long-term working state, and the excellent electrochemical properties of the sensor were key to performing precise measurements. In order to ensure the universality and adaptability of SCS monitoring, oysters belonging to three different grades (superior, medium, inferior) were selected as experimental objects.

#### 2.5.1. Sensor Electrochemical Performance Test 

The response of the flexible pressure sensor depends on the pressure applied to the sensitive area of the sensor. At no load, the reduced graphite oxide bulge is released. When the pressure is applied directly above, the voids in the reduced graphene oxide structure are reduced, and the contact between the inner carbon-containing structures is closer, thus improving the lateral conductivity of the inner electrons. This shows the electrical properties, where with increasing pressure, the convex structure is compressed, and the resistance decreases accordingly. In addition, sensitivity, repeatability, and response time are also important performance indicators of sensors. Firstly, a repeatable mechanical force system was used to apply pressure to the flexible pressure sensor. Next, electrochemical workstations were connected at both ends of the electrodes to record the electrical characteristics with pressure, as shown in Figure 6a.

#### 2.5.2. Oyster SCS Change Characteristics

Pacific oysters were harvested from a commercial farm in the fishing area of Rushan City, Shandong Province, China, in February 2022. Live oysters were packed in polyethylene bags and transported by refrigerated truck (4 °C) to the laboratory within 2 h. They were placed in a thermostat at 4 °C (experimental group 1) and 25 °C (experimental group 2) to measure SCS characteristics.

Step 1: Running water was used to rinse and remove dirt from the surface of the oyster shell. The oysters were divided into three groups on the basis of quality: superior (300 ± 10 g), medium (250 ± 10 g) and inferior (200 ± 10 g).

Step 2: The flexible pressure sensor was installed between the two shells after prying open a slit in the oyster. Each sensor device monitored the SCS changes in 12 oysters and uploaded data every 15 min. The oysters were stimulated every 1 h to ensure that their shells remained tightly closed. When oyster SCS remained 0 after stress, they were considered to be dead.

## 3. Results and Discussion

### 3.1. Electrical and Mechanical Test of Sensor

Figure 7a depicts the relationship between voltage and current at different pressures. It is obvious that the slope of the I-V curve decreases with increasing applied strain. According to Ohm’s law, the slope on the I-V curve corresponds to the resistance, which means that the resistance changes significantly with the applied strain. Figure 7b presents the results of the stability tests of the sensor at pressures of 5, 10, 20, 50, 100, and 140 kPa, respectively, with cyclic pressure application intervals of 10 s. $\Delta R/R_{0}$ represents the change in relative resistance.
(4)$$\frac{\Delta R}{R_{0}}=\frac{R-R_{0}}{R_{0}}$$
where R is the resistance value after applying pressure, and $R_{0}$ is the initial resistance value.

In the same batch of sensors, In this experiment, 10 pressure sensors were randomly selected for stability testing. The relative resistance responses of the sensors were essentially the same when the same pressure was applied, indicating that the sensor has excellent stability. As shown in Figure 7c, the sensor exhibited excellent durability and repeatability after 3000 repeated load/unload cycles at 150 KPa. As shown in the enlarged insets, almost all resistance response amplitudes were consistent after each load/unload pressure cycle, suggesting that the sensor has a long service life and high stability. Figure 7d shows the response time of the rGO-PDMS sensor for loading/unloading pressure. When the pressure on the resistance sensor is unloaded, the relative resistance value remains nearly constant. The relative resistance decreases immediately when external pressure is applied instantaneously, and its enlarged graph clearly shows that the response time of the sensor is about 0.3 s, as shown in Figure 7e. In summary, the piezoresistive flexible sensor based on rGO-PDMS demonstrates high stability and repeatability. In addition, the rGO-PDMS sensor does not require expensive micromachining tools or materials, and can be effectively applied for SCS monitoring during oyster transportation and storage.

### 3.2. Shell-Closing Strength Change

When the oysters were transported to the laboratory and subjected to external stress, such as due to vibration, temperature and light, the shells remained closed. As shown in Figure 8, in experimental group 1, the initial SCS of the three grades of oysters were 3318 g, 2756 g, and 2123 g, respectively; in experimental group 2, the initial SCS of the three grades of oysters were 3329 g, 2799 g, and 2089 g, respectively. The relationship between oyster grade and SCS was explored, and the results showed that the initial SCS values of superior oysters were greater. Further analysis indicated that there were significant differences in the SCS values of oysters from different grades (p<0.05). This is because the superior oysters had large adductor muscles, and the adductor muscle is the most direct factor in maintaining the tightness of the oyster shell [29].

Then, the relationship between SCS and survival time was discussed. In experimental group 1, the survival times of the superior, medium, and inferior oysters at 4 °C were 31, 25 and 18 days, and the survival time was significantly different (p<0.05). Meanwhile, in experimental group 2, the survival times of the superior, medium, and inferior oysters at 25 °C were 12, 10 and 7 days, and the survival time was significantly different (p<0.05). As shown in Figure 8c, superior oysters were more adaptable to environmental fluctuations and less affected by external stress [30]. In addition, this study further explored the effect of temperature on the survival time of oysters. Compared with at 25 °C, oysters survived longer at 4 °C, and temperature had a significant effect on survival time (p<0.05). Therefore, larger oysters are suitable for prolonged storage and transport over longer distances at lower temperatures. It is worth noting that a large temperature gradient interval was selected in this experiment, and whether 4 °C is the best survival temperature for oysters remains to be further explored. Trends in oyster SCS over time for different grades and at different temperatures are shown in Table 1.

Although SCS gradually decreased over time, it is worth noting that this decline was not linear. In terms of the change trend, the oyster SCS did not decline rapidly in the early stage, but rather had a delayed declining trend, as shown in Figure 9. The adductor muscle stores glycogen in autumn and winter, which is an important organ for energy storage and conversion during oyster reproduction and metabolic processes. Therefore, at the beginning, the oyster SCS was maintained by glycogen in the adductor muscle. Over time, in order to maintain their own metabolism, oysters gradually consume the glycogen originally stored in their body, resulting a gradual weakening of the SCS [29]. Figure 9 shows that the delayed decline in SCS is related to oyster size and temperature. For oysters of the same size, the delay in the decline at 4 °C was longer than that at 25 °C, because more glycogen is consumed to maintain metabolism at low temperatures [31]. At the same temperature, larger oysters had a longer delay, because larger oysters store more glycogen in order to maintain their metabolism.

### 3.3. Survival Rate Prediction

During the 32-day oyster storage phase, key indicators (SCS, temperature, grade) were used as model data sets. The ARIMA model was used to predict the live oyster status, and the output was the survival rate. This study adopted 80% of the dataset as the model training set and 20% of the experimental data for model diagnostic evaluation. The most important thing is to find the best-grade live oysters to achieve high-precision prediction. The accuracy of the survival rate predictive model is shown in Figure 10.

The survival rate prediction accuracies for superior, medium, and inferior live oysters at 4 °C were 89.32%, 82.17%, and 79.19%, respectively. The survival rate prediction accuracies for superior, medium, and inferior live oysters at 25 °C were 82.63%, 78.47%, and 72.93%, respectively. It can be clearly seen that the accuracy of the survival rate prediction is higher for superior oysters than for inferior oysters. Meanwhile, the prediction accuracy of for superior, medium, and inferior live survival rate of oysters at 4 °C was higher than that at 25 °C. The weakness of the inferior oysters in adapting to external stress is the main reason for the poor accuracy. These research results can serve as a reference for the diagnosis and prediction of the status of live oysters, but further optimization is still necessary to improve the accuracy of the model.

### 3.4. System Performance Evaluation

This system was able to effectively obtain data on SCS throughout the experiment, providing useful information regarding physiological changes in oysters. Oyster survival was effectively evaluated by monitoring SCS using a flexible pressure sensing system. An evaluation of the overall system performance is presented in Table 2. It can be seen that the monitoring system is non-destructive, the response is rapid, the traceability is strong, and the predicted survival rates of the oysters is accurate.

## 4. Conclusions

Understanding the physiological mechanisms and levels of stress responses during the storage and transportation of live oysters is key to maximizing survival time. This study developed a piezoresistive flexible sensor device based on rGO-PDMS. After testing the performance of the sensor, the device was used to investigate the SCS variation characteristics of three grades of live oysters at different temperatures. The results indicate that the rGO-PDMS flexible pressure sensor has the advantages of excellent electrochemical performance, low price, high stability, and good repeatability, and can be effectively used in the monitoring of physiological signals in live oysters.

At the same time, SCS is an important physiological index of live oysters, and temperature and size are the key factors affecting survival time. Dynamic capture of living physiological signals is of great significance in improving survival time during storage and transportation. This also provides a reliable and effective research basis for further exploring the relationship between physiological signals and quality.

However, exploring oyster physiological signals is a challenging field. Continuous exploration and development of new biosensors can more effectively improve the survival time of oysters, and promote the application of flexible sensing technology in other shellfish.

## Figures and Tables

**Figure 1 sensors-23-01308-f001:**
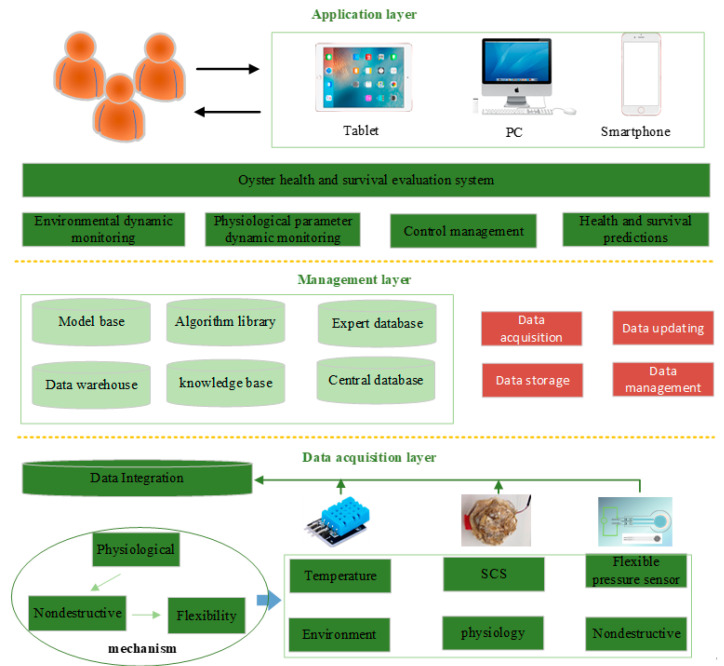
Overall framework of the intelligent system for oyster health evaluation.

**Figure 2 sensors-23-01308-f002:**
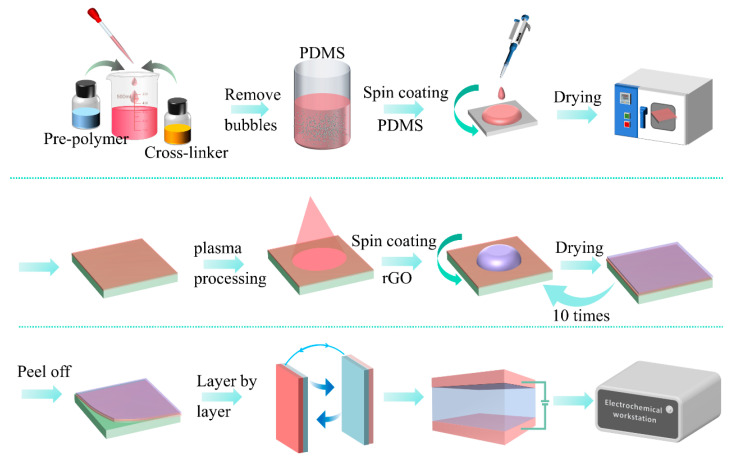
Schematic diagram of the fabrication process of the reduced graphene oxide-polydimethylsiloxane (rGO-PDMS) sensor.

**Figure 3 sensors-23-01308-f003:**
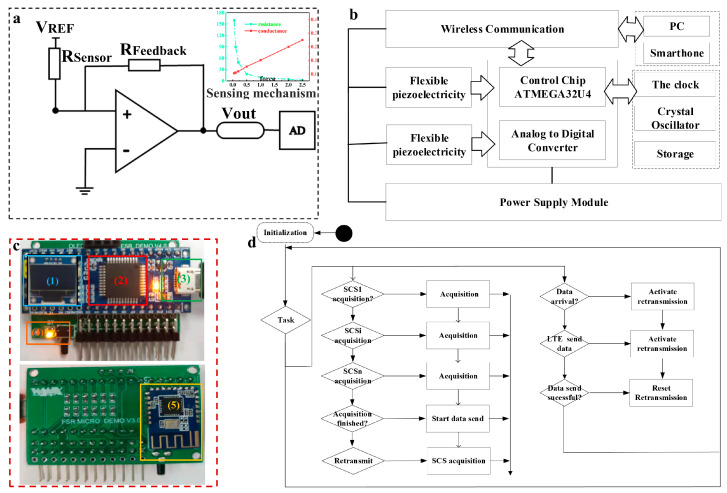
Flexible sensing system hardware design schematic diagram. (**a**) Sensor operation amplifier circuit; (**b**) the circuit block diagram of the sensor acquisition system; (**c**) sensor system hardware display, including: (1) real-time display; (2) ATMEGA32U4 chip; (3) universal serial bus (USB) serial port; (4) Bluetooth connection indicator; (4) light-emitting diode (LED) indicating lamp; (5) Bluetooth 4.0 module; (**d**) sensor signal processing process schematic diagram.

**Figure 4 sensors-23-01308-f004:**
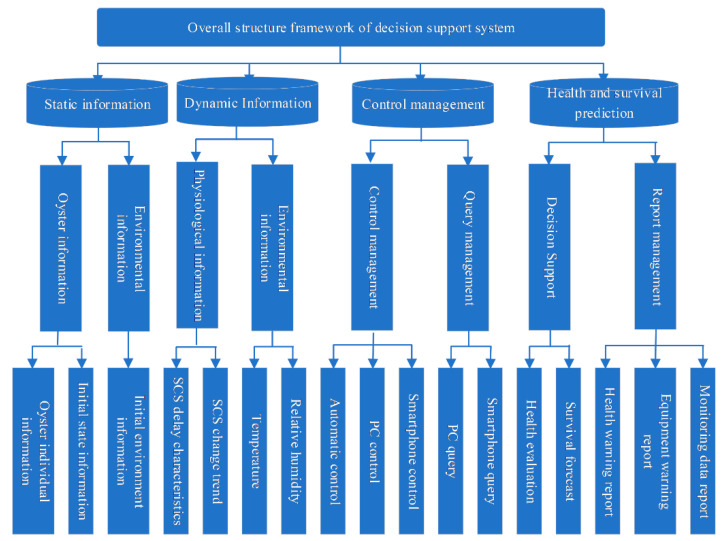
Schematic diagram of the overall architecture of the support system.

**Figure 5 sensors-23-01308-f005:**
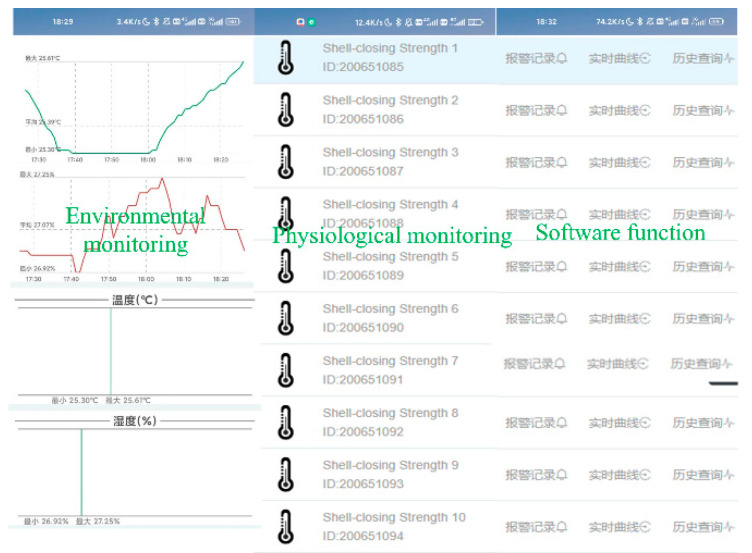
Software system monitoring interface display.

**Figure 6 sensors-23-01308-f006:**
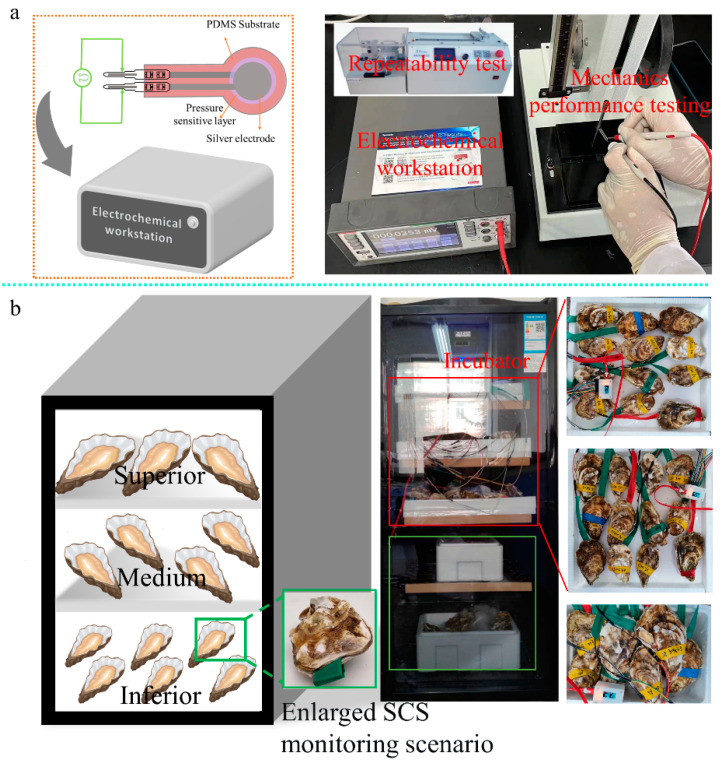
Experimental scenario and process design. (**a**) Sensor electrochemical performance test; (**b**) oyster shell-closing strength (SCS) change characteristic monitoring experiment.

**Figure 7 sensors-23-01308-f007:**
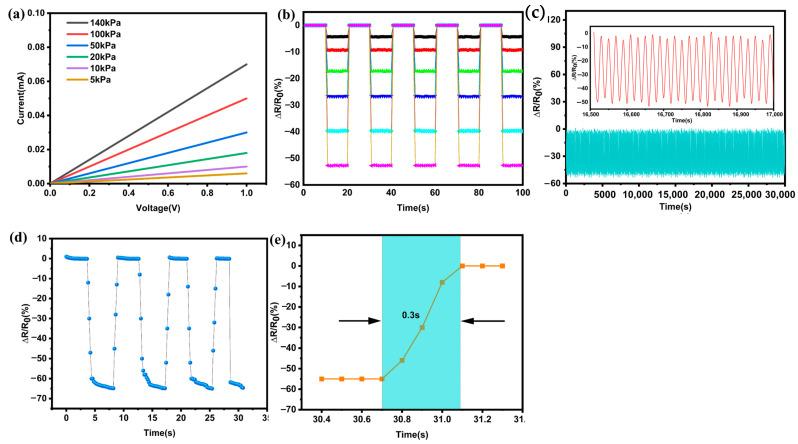
Electrical and mechanical test of the rGO-PDMS. (**a**) I-V curves of the rGO-PDMS with various amounts of applied pressure; (**b**) repeated cyclic tests of loading and unloading pressure at 5, 10, 20, 50, 100, and 140 kPa, respectively; (**c**) stability test of the rGO-PDMS sensor for 30,000 cycles of loading and unloading; (**d**) response time of rGO-PDMS sensor upon loading/unloading; (**e**) expanded view of rGO-PDMS sensor response time.

**Figure 8 sensors-23-01308-f008:**
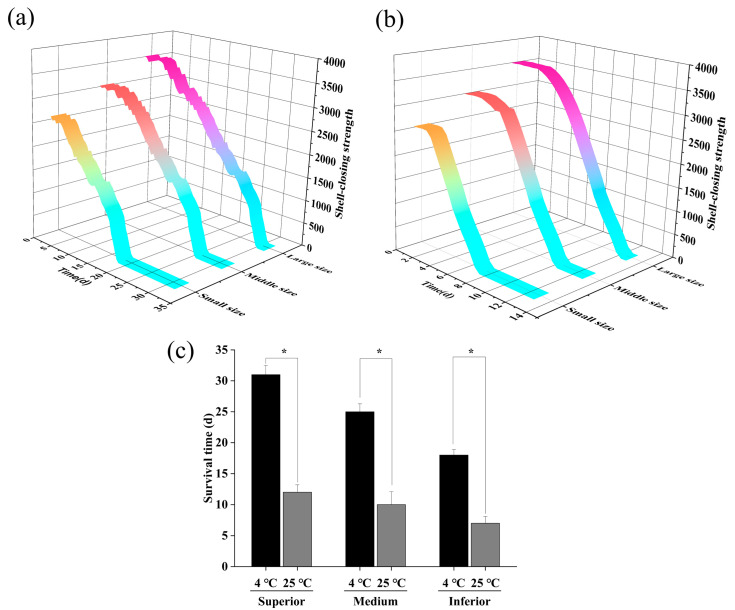
Oyster shell-closing strength change trend. (**a**) Live oyster SCS change characteristics under 4 °C; (**b**) live oyster SCS change characteristics under 25 °C; (**c**) comparative analysis of survival time at different temperatures. * Indicates significant difference in survival time of oysters at 4 °C and 25 °C; different letters indicate significant differences in survival time of oysters of different sizes at the same temperature.

**Figure 9 sensors-23-01308-f009:**
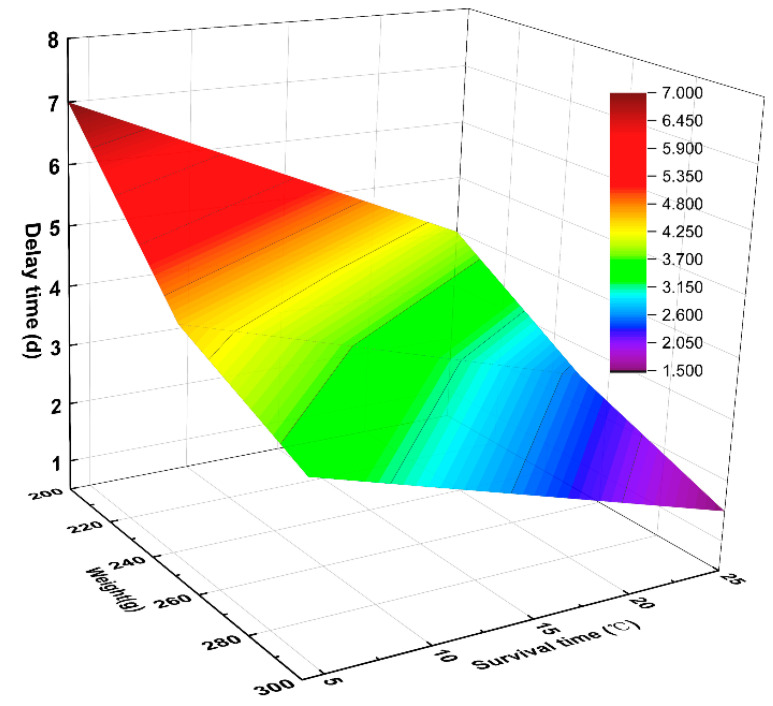
Analysis diagram of SCS delay characteristics.

**Figure 10 sensors-23-01308-f010:**
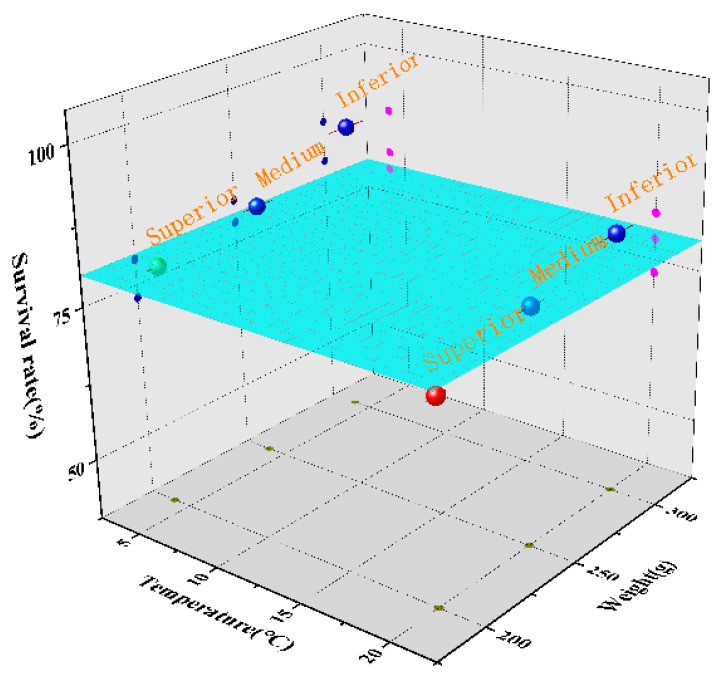
Prediction results for oyster survival.

**Table 1 sensors-23-01308-t001:** Trends in oyster SCS over time for different grades and at different temperatures (mean ± SD). $CV=\left(\frac{SD}{mean}\right)*100\%$; while different small letters indicate significant differences between time intervals within each group (P<0.05).

Grades	Temperature	Storage Time	SCS (g)	CV
Superior	4 °C	0	3511 ± 42.32 ^a^	0.012
		5	3502 ± 27.87 ^a^	0.008
		10	2850 ± 33.96 ^b^	0.012
		15	2432 ± 12.78 ^c^	0.005
		20	1801 ± 98.66 ^d^	0.055
		25	1201 ± 42.97 ^e^	0.036
		30	57 ± 11.53 ^f^	0.202
	25 °C	0	3493 ± 59.96 ^a^	0.017
		3	3452 ± 45.87 ^a^	0.013
		6	2825 ± 113.21 ^b^	0.040
		9	1112 ± 66.72 ^c^	0.060
		12	102 ± 12.84 ^e^	0.13
Medium	4 °C	0	3020 ± 56.65 ^a^	0.019
		5	2915 ± 69.75 ^a^	0.024
		10	2703 ± 78.37 ^a^	0.029
		15	1800 ± 32.13 ^b^	0.018
		20	1361 ± 45.38 ^c^	0.033
		25	233 ± 22.30 ^d^	0.096
	25 °C	0	2873 ± 33.67 ^a^	0.012
		3	2825 ± 45.96 ^a^	0.016
		6	1858 ± 26.59 ^b^	0.014
		9	326 ± 21.47 ^c^	0.066
Inferior	4 °C	0	2560 ± 69.69 ^a^	0.027
		5	2110 ± 53.82 ^c^	0.025
		10	1501 ± 52.75 ^c^	0.035
		15	980 ± 33.47 ^d^	0.034
	25 °C	0	2520 ± 36.55 ^a^	0.014
		3	1789 ± 55.21 ^b^	0.031
		6	523 ± 41.87 ^c^	0.080

Note: SD means standard deviation, CV means the coefficient of variation.

**Table 2 sensors-23-01308-t002:** Overall system performance.

System Performance	System Platform Real-Time Monitoring Performance			Model Application Performance		
				Superior	Medium	Inferior
Measure Parameters	Temperature	Relative Humidity	Shell-Closing Strength	Survival Rate	Survival Rate	Survival Rate
Traditional systems	Accuracy: 0.4% Range: 40~80 °C	Accuracy: 0.4% Range: 40~80 °C	Destructive measurement	Unable to survive forecast		
Current system	-	-	Flexible non-destructive measurement	Accuracy: 89.32%	Accuracy: 82.17%	Accuracy: 79.19%
advantages	Real -time monitoring, remote control, data records, higher traceability		non-destructive, sustainable monitoring	More conducive to the survival management of oysters, improve the survival rate and economic benefits		

## Data Availability

The data presented in this study are available on request from the corresponding author.

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
