# Peer review of "rGO-PDMS Flexible Sensors Enabled Survival Decision System for Live Oysters"

_sensors, 2023, doi:10.3390/s23031308_

Round 1

Reviewer 1 Report

The scientific results from this paper give the contribution in digital supply chain of the food industry as well-known important sector for quality of life.

Well done analysis verified and validate by cutting-edge scientific based methods and techniques.

Author Response

Dear Editors and Reviewers:

Thank you very much for your recognition!

Reviewer 2 Report

In this work, the authors reported a strategy to predict the health status of the oysters by monitoring their SCS by rGO-PDMS Flexible Sensors. The work was systematacially finished, but the manuscript was not well prepared. We have the following suggestions before its publication.

1.      What is LIG blending sensor? In page 2/14

2.      Why the flexibility of the sensors is necessary for monitoring the healthy status of oysters? Can the traditional rigid sensors also work for this application? Compared with flexible sensors, the traditional rigid sensors show higher stability and repeatability.

3.      The authors are suggested to emphasize the novelty of this work.

4.      There are some misleading of the figures, for example, in the section” 3.1. Electrical and mechanical test of sensor”, Figure 4 should be Figure 7.

5.      How about the repeatability of the sensors prepared in this work?

Author Response

Dear Editors and Reviewers:

        Thank you for your letter and for the reviewers’ comments concerning our manuscript entitled “rGO-PDMS Flexible Sensors enabled Survival Decision System of Live Oysters”. Those comments are all valuable and very helpful for revising and improving our paper, as well as the important guiding significance to our researches. We have studied comments carefully and have made correction which we hope meet with approval. The main corrections in the paper and the responds to the reviewer’s comments are as following:

1.“What is LIG blending sensor? In page 2/14” LIG has been modified to laser-induced graphene.

2.“Why the flexibility of the sensors is necessary for monitoring the healthy status of oysters? Can the traditional rigid sensors also work for this application? Compared with flexible sensors, the traditional rigid sensors show higher stability and repeatability.”

Answer:

The traditional rigid pressure sensor has a large thickness, which will have the following defects when monitoring the shell-closing strength:

(1). Traditional rigid sensors have a large volume, which is not conducive to installing between double shells when monitoring SCS.

(2). Oysters open and close their shells as they metabolize and breathe normally. The traditional sensor has a large thickness, which will directly affect the normal metabolism of live oysters, directly affect the health of oysters.

(3). The shell of a live oyster remains closed when exposed to external pressure. If we use conventional sensors, oyster shells can't be tightly closed while monitoring. At this point, the adductor muscles in the living oyster remain tense. This not only causes damage to the adductor muscles, but also affects the oyster's health.

Therefore, although the traditional sensor has better stability and repeatability, we can not use this traditional sensor in order to ensure the health of oysters. The flexible sensors are less stable and repeatable than the rigid ones, but live oysters have a limited life span (about 30 days in optimum conditions). During this time period, flexible pressure can also work effectively. Therefore, the choice of flexible sensor has greater advantages.

3.“The authors are suggested to emphasize the novelty of this work.” It has been added to the paper to emphasize novelty.

4.“There are some misleading of the figures, for example, in the section” 3.1. Electrical and mechanical test of sensor”, Figure 4 should be Figure 7.” It has been modified in the paper.

5.“How about the repeatability of the sensors prepared in this work” We conducted 3000 repeatability tests. In the repeatability test, we apply a steady pressure and record the sensor response. The results show that the sensor has good repeatability and can be used to monitor live oyster shell closing strength.

Reviewer 3 Report

In this paper, a flexible sensor based on rGO-PDMS was developed to investigate physiological mechanisms for oysters. I enjoyed reading the manuscript and recommend it to be published in Materials. However, prior to acceptance, authors should address the following comments:

1. In Introduction, “In addition, PDMS-coated LIG…”, the full name for “LIG”, which is supposed to be laser induced graphene is needed.

2. In Introduction, “Previous literature indicated that the closure strength was the most important index to evaluate the health status of oyster.” References are needed here.

3. In 2.2, “PDMS was spin-coated (1000 rpm) onto the glass slide surface…”. The time for spin-coating is needed here.

4. In 2.2, “The dried PDMS film was plasma processed for 10 min to enhance its hydrophobicity.”. After plasma processing, hydrophilicity not hydrophobicity is enhanced.

5. In 2.2, “… then repeatedly spin-coated with rGO solution 10 times (r = 600 rpm).” The time for spin-coating is needed here. Moreover, the information for rGO, like concentration, what solution based… are needed here.

6. In Figure 2, after layer by layer, a green color appeared in the middle, what does that stand for? I suggest labelling the name for different colors.

7. In 2.3.1, The equation number is needed.

8. In Figure 3, the font for different panels should be consistent.

9. In Figure 4, the most top square is blank.

10. In 2.5.1, the sensing mechanism is supposed to be discussed, like under pressure, rGO flakes move, then the contact resistance and material resistance change.

11. In 2.5.2, “transport ed” should be “transported”.

12. In 2.5.2, “Step 1: Use running water to rinse and remove…” should be “Step 1: Running water was used to rinse and remove…”.

13. In 2.5.2, “When oyster SCS remained 0 after stress, they were considered dead.” How did authors calculate SCS values? In addition, how much is the variation for different sensors?

14. In Figure 6b, what does the green frame mean? I suggest labelling as much information as you can in these photos.

15. In 3.1, “The sensor resistance response is essentially the same when the same pressure is applied.” What does resistance response mean, absolute resistance change or relative resistance change? How many sensors were used when the authors drew this conclusion?

16. In Figure 7a, how did authors measure the current and what did current and voltage stand for?

17. In Figure 7b, how did authors calculate DR?

18. In Figure 7, why did the curves have plateau in Figure 7b but not in Figure 7c?

19. In 3.1, “As shown in Figure 4(c)… Figure 4(d) shows… as shown in Figure 4(e)” should be “As shown in Figure 7(c)… Figure 7(d) shows… as shown in Figure 7(e)”.

20. In Figure (c), the time needed for 3000 cycles is 2750 s, however, the authors mentioned “with cyclic pressure applications intervals of 10s”. Then I suppose 30000 s is needed for 3000 cycles.

21. In Figure 7, why did the curves have a constant plateau in Figure 7b but a changing plateau in Figure 7d?

22. In 3.2, what did P < 0.05 mean?

23. In 3.2, “As shown in Figure 8(c), superior oysters were more adaptable to environmental fluctuations and less affected by external stress.” What is the data that can support this statement?

24. In Table 1, how did authors calculate CV values?

25. In 3.2, “…figure 9” should be “…Figure 9”.

26. In Figure 9, how did authors measure delay time, what is the unit for delay time and what is the unit for survival time (I am sure it cannot be °C)?

27. In Table 2, reference for traditional system is needed. “advantages” should be “Advantages”. How did Relative humidity range from 40 °C to 80 °C? “non-destructive” should be “Non-destructive”. Please also add standard deviations to accuracy.

28. In Conclusion, “…and promote the promotion and application of flexible sensing technology to other shellfish sensing technology to other shellfish”. This sentence is difficult to be understood.

Author Response

Dear Editors and Reviewers:

        Thank you for your letter and for the reviewers’ comments concerning our manuscript entitled “rGO-PDMS Flexible Sensors enabled Survival Decision System of Live Oysters”. Those comments are all valuable and very helpful for revising and improving our paper, as well as the important guiding significance to our researches. We have studied comments carefully and have made correction which we hope meet with approval. The main corrections in the paper and the responds to the reviewer’s comments are as following:

  1. In Introduction, LIG has been modified to laser-induced graphene.
  2. In Introduction, “Previous literature indicated that the closure strength was the most important index to evaluate the health status of oyster.” Here I have added references
  3. In 2.2, “PDMS was spin-coated (1000 rpm) onto the glass slide surface…”. I have added spin-coating time
  4. In 2.2, “The dried PDMS film was plasma processed for 10 min to enhance its hydrophobicity.”. After plasma processing, hydrophilicity not hydrophobicity is enhanced. According to your suggestion, I have modified it in the paper
  5. In 2.2, “… then repeatedly spin-coated with rGO solution 10 times (r = 600 rpm).” I added in the paper spin-coating time, concentration and solution based on N, N-dimethylformamide.
  6. “In Figure 2, after layer by layer, a green color appeared in the middle, what does that stand for?” I deleted the green here and explained.
  7. “In 2.3.1, The equation number is needed.” I've numbered the equation here.
  8. “In Figure 3, the font for different panels should be consistent.” The fonts in Figure 3 have been processed in the paper to make them consistent.
  9. “In Figure 4, the most top square is blank.” The white space has been filled with font
  10. “In 2.5.1, the sensing mechanism is supposed to be discussed, like under pressure, rGO flakes move, then the contact resistance and material resistance change.’’ In the paper, we have discussed the mechanism of the sensor.
  11. “In 2.5.2, “transport ed” should be “transported”.” It has been modified here
  12. “In 2.5.2, “Step 1: Use running water to rinse and remove…” should be “Step 1: Running water was used to rinse and remove…”.” It has been modified here
  13. “13. In 2.5.2, “When oyster SCS remained 0 after stress, they were considered dead.” How did authors calculate SCS values? In addition, how much is the variation for different sensors?” Explanation: In this paper, SCS (g) can be directly displayed on the system interface after digital to analog conversion of the sensor. With the decrease of oyster vitality, SCS will gradually become smaller. After the death of oyster, the closed shell force was no longer generated, and the SCS became 0.
  14. “In Figure 6b, what does the green frame mean? I suggest labelling as much information as you can in these photos.” The contents of the green frame are explained
  15. “In 3.1, “The sensor resistance response is essentially the same when the same pressure is applied.” What does resistance response mean, absolute resistance change or relative resistance change? How many sensors were used when the authors drew this conclusion?” In the same batch of sensors, we randomly selected 10 pressure sensors for stability testing.
  16. “16. In Figure 7a, how did authors measure the current and what did current and voltage stand for?” Explanation: I-V curve is mainly used to better represent the change of resistance. When the pressure is applied, the given voltage value will get a current through the sensor, using a multimeter to measure.
  17. “In Figure 7b, how did authors calculate DR?” In the word document of the paper, we have modified this problem.
  18. “In Figure 7, why did the curves have plateau in Figure 7b but not in Figure 7c?” Explanation: Figure 7(b) is the stability test, and Figure 7(D) is the response time test
  19. “In 3.1, “As shown in Figure 4(c)… Figure 4(d) shows… as shown in Figure 4(e)” should be “As shown in Figure 7(c)… Figure 7(d) shows… as shown in Figure 7(e)”.” This has been modified.
  20. “In Figure (c), the time needed for 3000 cycles is 2750 s, however, the authors mentioned “with cyclic pressure applications intervals of 10s”. Then I suppose 30000 s is needed for 3000 cycles. ”The time marking error in Figure 7 (c) has been corrected.
  21. “In Figure 7, why did the curves have a constant plateau in Figure 7b but a changing plateau in Figure 7d?” Explanation: Figure 7(b) is the stability test, and Figure 7(d) is the response time test
  22. “In 3.2, what did P < 0.05 mean?” In statistics, (P<0.05) means that there are significant differences between samples.
  23. “In 3.2, “As shown in Figure 8(c), superior oysters were more adaptable to environmental fluctuations and less affected by external stress.” What is the data that can support this statement?” There are similar explorations in the references [29][30]. In our own experiments, we stored different grades of oysters at the same temperature (4 ° C and 25 ° C), and the results showed that high-quality oysters survived longer, which was verified by experiments.
  24. “In Table 1, how did authors calculate CV values?” Explanations have been added to the paper. CV= (Standard Deviation/mean) x 100%.
  25. “In 3.2, “…figure 9” should be “…Figure 9”.” It has been modified in the paper
  26. “In Figure 9, how did authors measure delay time, what is the unit for delay time and what is the unit for survival time (I am sure it cannot be °C)?” The delay time is in days and has been modified in Figure 9.
  27. “In Table 2, reference for traditional system is needed. “advantages” should be “Advantages”. How did Relative humidity range from 40 °C to 80 °C? “non-destructive” should be “Non-destructive”. Please also add standard deviations to accuracy.” Accuracy and range are for traditional sensors. An error occurred while filling in the form, which has now been corrected.
  28. “In Conclusion, “…and promote the promotion and application of flexible sensing technology to other shellfish sensing technology to other shellfish”. This sentence is difficult to be understood.” The statement has been remodified.